# Instance Segmentation with Mask R-CNN Applied to Loose-Housed Dairy Cows in a Multi-Camera Setting

**DOI:** 10.3390/ani10122402

**Published:** 2020-12-15

**Authors:** Jennifer Salau, Joachim Krieter

**Affiliations:** Institute of Animal Breeding and Husbandry, Kiel University, Olshausenstraße 40, 24098 Kiel, Germany; jkrieter@tierzucht.uni-kiel.de

**Keywords:** machine learning, Mask-R-convolutional neural networks, dairy cattle, multi-camera video surveillance, object recognition

## Abstract

**Simple Summary:**

As sociability in cattle is a matter of good animal husbandry, this study provides technical groundwork for a camera based system to automatically analyse dairy cattle herd activity. Eight surveillance cameras were recording a group of thirty-six lactating Holstein Friesian dairy cows. A Mask R-CNN model was trained to determine pixel level segmentation masks for the cows in the video material. The animals were successfully segmented reaching high ‘averaged precision scores’ for bounding boxes (0.91) and segmentation masks (0.85) for a given IOU threshold of 0.5. As providing training data for deep learning models is time consuming and tedious, this article also deals with the question “How many images do I have to annotate?” and analyses the performance of the model depending on the size of the used training data set.

**Abstract:**

With increasing herd sizes came an enhanced requirement for automated systems to support the farmers in the monitoring of the health and welfare status of their livestock. Cattle are a highly sociable species, and the herd structure has important impact on the animal welfare. As the behaviour of the animals and their social interactions can be influenced by the presence of a human observer, a camera based system that automatically detects the animals would be beneficial to analyse dairy cattle herd activity. In the present study, eight surveillance cameras were mounted above the barn area of a group of thirty-six lactating Holstein Friesian dairy cows at the Chamber of Agriculture in Futterkamp in Northern Germany. With Mask R-CNN, a state-of-the-art model of convolutional neural networks was trained to determine pixel level segmentation masks for the cows in the video material. The model was pre-trained on the Microsoft common objects in the context data set, and transfer learning was carried out on annotated image material from the recordings as training data set. In addition, the relationship between the size of the used training data set and the performance on the model after transfer learning was analysed. The trained model achieved averaged precision (Intersection over union, IOU = 0.5) 91% and 85% for the detection of bounding boxes and segmentation masks of the cows, respectively, thereby laying a solid technical basis for an automated analysis of herd activity and the use of resources in loose-housing.

## 1. Introduction

Through the field of Precision Livestock Farming (PLF), the use of sensors and cameras as well as machine learning and image processing techniques have entered the agricultural sciences. As the world population is growing rapidly, PLF is addressing the problem to provide sustainable food production [1]. As herd sizes are constantly increasing, farmers benefit from technical solutions to support them in monitoring the health- or welfare status of their animals [2,3]. Cameras are a non-invasive method of collecting data from animals, and a lot of camera-based studies have been carried out in PLF during the past years. Hereby, various types of cameras were applied with regard to a wide range of topics. 2D video cameras [4,5] as well as 3D depth cameras [6,7,8,9] started to be used in the detection of lameness. The problem of body condition determination was successfully approached by [10] using thermal cameras, but also in 2D camera projects [11] and especially 3D camera projects [12,13,14,15]. Recent topics of camera based studies were the monitoring of herd activities [16], conformation recording [17], animal identification [18], animal behaviour [19,20] and the animals’ use of barn space [21].

In the present article, eight surveillance cameras were used to record a group of lactating dairy cows in a loose-housing barn. To monitor the herd behaviour, the setting was designed so that the animals could not leave the monitored area. As cattle naturally live in herds and have a strong need for companionship, dairy cows were usually kept in groups. Cattle form cliques and prefer acting together, thus, loose-housing in groups could reduce stress when the animals were given the opportunities to express their natural behaviour patterns. Ref. [22] showed that the social structure in herds of semi-wild cattle was based on matriarchal families, whilst the behaviour of the herd and the social structure are influenced by the individuals in the herd [23,24]. Ref. [25] stated that the number of animals in the herd was no reliable indicator of the on-farm animal welfare level, but housing and management influence the welfare status more than the herd size. It is furthermore known that the expression of social behaviour and herd activity influence the health [26] and productivity [27] of dairy cattle. Therefore, a deep knowledge on the group activities in dairy cattle is a matter of animal welfare and good husbandry, and there has been a rising number of studies on this topic. Šárová et al. analysed the synchronisation of herd activity from visual observations [28], and Nelson et al. combined visual observation with positioning sensors [29] for activity monitoring and estrus detection. Often, data from the global positioning systems (GPS), indoor positioning systems (for example Ubisense) or by proximity loggers have been applied to study herd activities. GPS sensors were used by Davis et al. to analyse travelling distances [30]. In [31], walking behaviour was dealt with using an Ubisense indoor positioning system, and Boyland et al. used proximity loggers to analyse social networks and group structure [32]. However, the application of wireless sensors in animal related studies has some disadvantages. Firstly, it could not be totally excluded that the animals are disturbed by the attached sensors. Additionally, the wireless sensors are prone to destruction by the animals as well as interferences with humidity or metal, as shown in [33]. Furthermore, in [30], it was proven that the necessary trade-off between data sampling frequency and battery lifetime could significantly affect the results of the study. It is, therefore, a downside of camera data that the animal related information needs to be calculated from the image or video material using sophisticated image processing techniques and machine learning for automation.

Artificial neural networks [34] originated from the attempt to model the neural activity of the human nerve system [35]. Similar to the human brain, artificial neural networks are able to learn underlying pattern through repeated processing of numeric data. In addition to fully connected neural networks, Ref. [36] introduced the first convolutional neural network (CNN) in which some of the fully connected layers were replaced by convolutional layers. This means that the input to the layer is convolved with a sliding dot product [37]. Due to high computational costs, the breakthrough of CNN applications happened not before it was possible to implement them on graphical processing units (GPU) to accelerate the training [38,39]. CNNs also came to use in agricultural sciences, as Ref. [40] compared several neural network models trained on motion data of steers to distinguish between behaviours like feeding, lying, ruminating, licking salt, moving, social licking and head butt. However, mostly CNNs were used in the context of image processing. In [41], CNNs were trained on depth images to estimate body condition.

The present article uses CNNs to detect dairy cows in the video material. The topic of object recognition applied to freely moving animals is challenging, as the setting and the orientation of the animals could only be controlled to a certain extent, settings often suffer from varying light conditions and farm animals are comparably large object, leading to the problem that the field of view of one camera is usually not sufficient to capture the group. Animal recognition has been approached by [42] who successfully trained CNNs on the images taken by multiple time lapse cameras to specify the location of goats on a bounded pasture area. Multi-camera systems for the detection of dairy cows that are free to move around have been presented in [43,44]. While [43] used a Viola-Jones algorithm on a group of fifteen dairy cows in their freestall barn, in [44], CNNs were trained to detect cows in the waiting arena of an automatic milking system. The training of CNNs requires a large amount of high quality training data [45], which is often very time consuming to generate. There exist several openly available data sets provided by computer vision challenges like the PASCAL VOC (PASCAL Visual Object Classes [46]), ILSVRC (ImageNet Large Scale Visual Recognition Challenge [47]) and MS COCO (Microsoft Common Objects in Context [48]), which can be used for the training of CNN models. These data sets hold a large amount of labelled or annotated images featuring multiple classes of every day life objects like ‘person’, ‘house’, ‘car’, etc. In addition, models with outstanding performance—partly the winning models of the aforementioned challenges—concerning the detection of several classes of objects like SSD and YOLO9000 are available [49,50].

In the present article, a Mask R-CNN model (Mask region-based CNN [51]), which had been pre-trained on the MS COCO data set [52], was used to determine cows in the video material. Mask R-CNN is an extension of a state-of-the-art CNN object detection model Faster R-CNN (Faster region-based CNN [53]) with the speciality not only to deliver bounding boxes for the objects of interest, but pixel level segmentation masks. This is called instance segmentation, and it requires not only the correct detection of all objects, but also a precise segmenting of each instance. In a recently published article by [54], the superiority of Mask R-CNN compared to other state-of-the-art models was confirmed when applied to counting grazing cattle. In contrast to the approach presented here, Ref. [54] used the video material of drones to detect cattle in an outdoor environment. Even if the development of an automated system can be based on a pre-trained model, especially the indoor recording settings of agricultural science applications are often uncommon compared to the data set on which the model was pre-trained. It is, thus, beneficial for transfer learning to generate a training data set from the respective setting.

The objective of this article is to provide a working instance segmentation for a complete surveillance of a group of dairy cows in an indoor loose-housing setting. For many applications regarding an automated analysis of herd activities, the tracking of animals in video material is essential, and the presented detection of individual instances of cows establishes important technical groundwork. However, the instance segmentation itself could already be used for valuable analyses, as was exemplarily illustrated in this article with a visualisation of space usage based on Mask R-CNN instance segmentation. The object detection evaluation metrics averaged precision and averaged recall introduced in [55] are used to evaluate the performance of our trained Mask R-CNN model and highlight the advances compared to the pre-trained version. Furthermore, it was the goal to present an answer to the question, how many images derived from the setting under analysis need to be prepared for transfer learning, as the annotation of a lot of images is time consuming and tedious. Thus, it is additionally evaluated in this study how the number of self annotated images is reflecting on the performance of the model after transfer learning.

## 2. Materials and Methods

### 2.1. Hardware and Recording Software

Eight AXIS M3046-V internet protocol (IP) dome cameras by Axis Communications, Sweden with 1/3″ progressive scan RGB CMOS image sensor were used. Video files in mp4 format were delivered with resolution 1920 × 1080 pixels, 2 frames s${}^{-1}$ and the AXIS specific compression parameter was set to zero [56]. The 2.4 mm focal length lens provided 128${}^{\circ }$ horizontal field of view (FOV) and 72${}^{\circ }$ vertical FOV. The cameras were assigned fixed IP addresses and passwords for the protection of data and the privacy of personal as well as the in-house processes of the participating farm and powered by PoE. Recording software implemented in Python language [57] using the Open application programming interfaces (API) for software integration provided by AXIS (VAPIX^®^ [58]). For this application, the open VAPIX^®^ VERSION 3 Video Streaming API (www.axis.com) was used [58].

An Intel i5-8600 hexa core CPU clocked at 3.1 GHz on an ASUS CSM Corporate Stable Prime B360-Plus mainboard and 16 GB RAM (Samsung DDR4-2400) was used for camera operation, recording and data handling. This machine was later on equipped with a graphical processing unit (ASUS 8GB D6 RTX 2080S Turbo Evo) and used to train the convolutional neural networks (Section 2.3).

### 2.2. Data Collection

#### 2.2.1. Camera Installation and Recording

The rectangular barn had dimensions 26 m × 12 m and was equipped with thirty-six lying cubicles, eighteen feeding and two water troughs. The rubber floor was cleaned by an automated scraper. The cameras were installed above the centre line at 3.5 m height and placed ≈3 m apart. The devices were allocated ‘Camera 1’ to ‘Camera 8’. Their combined fields of view covered the complete barn. A schematic representation of the barn and the positions of the cameras could be found in Figure 1.

Between 3 April and 11 September 2019 footage was taken daily from 8:00 a.m. to 2:00 p.m. on the research farm Futterkamp of the Chamber of Agriculture in Northern Germany. This was the time between the milkings, thus all cows were present in the barn during recording and daylight conditions were ensured. It would have been beneficial to record for longer daily periods, but a privacy statement was necessary due to the General Data Protection Regulation, and recordings outside this time window were excluded. Additionally, the amount of data produced by eight continuously running cameras would have led to an overload in data traffic. Data were mirrored and stored between recording periods and removed from the recording system.

#### 2.2.2. Recorded Cows

The recorded group at the Lehr- und Versuchszentrum Futterkamp of the Chamber of Agriculture Schleswig-Holstein consisted of thirty-six Holstein Friesian dairy cows. Cows were fed ad libitum and fresh feed was provided after each milking. Cows were milked twice a day in a 2 × 12 side-by-side milking parlour. The recorded group consisted of 36 Holstein Friesian dairy cows and was assembled on 3 April 2019. The animals were put together from a total number of approximately 160 lactating Holstein Friesian cows held on the research farm in three groups. On 15 May and 17 June, respectively, six and four animals left the group due to drying off and were replaced by animals with similar milk yield and lactation number. Between 3 April and 15 May, cows were in the first to eight lactation (median = 2), and daily averaged milk yields varied from 24.7 kg to 48.0 kg (35.5 kg ± 5.9 kg). Between 15 May and 17 June, cows were in the first to eight lactation (median = 2), and daily averaged milk yields ranged from 22.3 kg to 50.1 kg (36.2 kg ± 5.8 kg). Since 17 June, the recorded cows were in the first to eight lactation (median = 2), and daily averaged milk yields ranged from 20.5 kg to 47.7 kg (37.1 kg ± 5.8 kg). Due to these exchanges, there were eleven to twelve primiparous cows, ten to eleven cows in second lactation, eight to ten cows in third lactation, two to three cows in fourth lactation, one to two cows in fifth lactation, and one cow in eighth lactation.

##### Ethical Statement

The disturbance of the animals is kept to a minimum as the study was designed to fit well into the normal routine of the farm. The authors declare that the “German Animal Welfare Act” (German designation: TierSchG) and the “German Order for the Protection of Animals used for Experimental Purposes and other Scientific Purposes” (German designation: TierSchVersV) were applied. No pain, suffering or injury was inflicted on the animals during the experiment.

### 2.3. Mask R-CNN

The Mask R-CNN model extends Faster R-CNN (Faster region-based CNN [53]) by providing instance segmentation in addition to object detection and classification. The model coincides with Faster R-CNN with regard to the backbone CNN and the Region Proposal Network (Section 2.3.1). While Faster R-CNN then trains two output branches, Mask R-CNN is additionally trained to predict pixel-to-pixel segmentation masks.

#### 2.3.1. Building Blocks of the Model

##### Backbone

The backbone of the Mask R-CNN implementation used in this study (Matterport [52]) is a ResNet50 or ResNet101 (Residual Network) with additional Feature Pyramid Network (FPN). In this application, ResNet50 was chosen. The backbone extracts features from the input image, starting with low level features like edges and corners specified by early layers, and in the later layers successively higher level features are detected.

ResNet50 is a deep CNN introduced by [59]. The architecture of ResNet50 comprises five stages. The first stage is composed of zero padding, a convolutional layer, batch normalisation [60], Rectified Linear Unit (ReLu) activation [61] and a max pooling layer. The later stages consist of convolutional layers followed by batch normalisation and ReLu activation. As a speciality of ResNet50/101, so-called skip connections are inserted in stages 2 to 4, meaning that the original input from the first layer of the respective stage is added to the output of the stacked convolutional layers.

Processing an image with the stacked convolutional layers of ResNet50 results in a pyramid of feature maps (bottom-up pathway, Figure 2) with decreasing spatial resolution but increasing semantic value as the number of convolutional steps rises. FPN was introduced by [62] and processes the output layers of the five ResNet50 stages further in an additional top-down pathway to construct higher resoluted layers from the semantically rich layers. With this, lower and higher level features can be accessed at every stage which improves the representation of objects at multiple scales.

##### Region Proposal Network (RPN)

To generate regions of interest (ROI), the feature maps generated from the ResNet50+FPN backbone are passed on to an RPN which determines the objectness, i.e., whether an object is present in an image region or not, and a bounding box in case. These regions are called anchors [53], and about 200,000 overlapping anchors per image are used. For the anchors that are most likely to contain an object, location and size are refined. A non-max Suppression is applied to remove anchors that overlap too much.

##### Three Output Branches

The Mask R-CNN model provides three outputs for every proposal from the RPN.
Classifier branch: As the ROIs suggested by the RPN can come with different sizes, a ROI pooling step is carried out to provide a fixed size for the softmax classifier [63]. The classifier is a region-based object detection CNN [64] which gives a confidence score con_sc∈(0,1), evaluating how likely the found object was a cow. In a multi class setting, confidence scores for all class labels would be output and a found object would be assigned to the most likely class. Additionally, a background class is generated and respective ROIs are discarded. As confidence threshold 0.9 was set, i.e., at all anchors with confidence score 0.9 or higher a cow was said to be present.Bounding box branch: To refine the bounding boxes and to reduce localization errors, an additional regression model is trained to correct the bounding boxes proposed by RPN [64].Instance segmentation branch: Masks of size 28 × 28 pixels are generated from the RPN proposals by an additional CNN. The masks hold float numbers, i.e., more information is stored than in binary masks, and they are later scaled to the size of the respective bounding box to receive pixel wise masks for the detected objects.

#### 2.3.2. Implementation and Training of Mask R-CNN

The model trained in the present article was based on the Mask R-CNN implementation by Matterport [52] using the Keras application programming interface [65]. As requirements, Python 3.4 [66], TensorFlow 1.3 [67] and Keras 2.0.8 were named. Since TensorFlow 2 was already available, the given implementation was manually upgraded to run with Python 3.6, TensorFlow 2.0 and Keras 2.3.1.

##### AxisCow

The model was pre-trained on the MS COCO data set [48,55], and was used with transfer learning, i.e., the weights from the former training were used as starting values. To further train the model for the more specific situation to find cows in the given recording setting (Section 2.2), the reading of training data and the model configurations needed to be adapted. The classes ‘AxisCowDataset’ and ‘AxisCowConfig’ inherited from the classes ‘utils.Datset’ and ‘Config’ defined in the Matterport implementation. Functions to read the data set as described in Section 2.4 were modified or added to ‘AxisCowDataset’, and configuration values were adapted.

##### Learning Rate Scheduler and Data Augmentation

The Mask R-CNN model was enhanced manually with a controlled decrease of the learning rate and data augmentation applied to the training data.

Decreasing the learning rate during training is based on the idea that a high learning rate enables a faster approach to the minimum of the loss function; however, after some epochs, refining the changes in the parameter could bring benefits in settling in the approached minimum. A learning rate scheduler was defined to lower the learning rate linearly with every epoch during training starting with a 0.01 initial learning rate. The scheduler was added to the Keras callbacks.

In the data augmentation step, the current batch of images were run through a pipeline of transformations prior to being fed to the model for training. In this way, the model is trained with slightly different information in every step. The Python module ‘imgaug’ [68] was loaded and four augmentation techniques were implemented. All augmentors were applied to a random 50% of the images in a batch and carried out in random order:A choice of two out of the following transformations: Identity, horizontal flip, vertical flip and cropping.The application of Gaussian blur or Gaussian noise.A change of contrast or a change of brightness.The application of an affine transformation.

##### Training

As the model was already pre-trained, here only stages four and five of the ResNet50+FPN backbone and all following parts of the model were trained. The training was run for 350 epochs.

### 2.4. Training and Validation Data Sets

For the training and validation data set 496, respectively, 104 images were randomly chosen from the recording days 6 April, 6 May, 12 June, 9 July, 5 August, and 3 September. These images were drawn from the videos and manually annotated by one person (Figure 3) using the free online tool VGG Image Annotator version 3 (VIA 3 [69,70]). In case an image contained no cow, it was skipped. This resulted in 479 images in the training data set and 96 images in the validation data set. The final numbers of annotated images were approximately the same for the eight cameras (Table 1).

The annotations were stored in JSON files (www.JSON.org) structured with the first order keys ‘project’, ‘config’, ‘attribute’, ‘file’, ‘metadata’, ‘view’. For every polygon, a dictionary with the image identification (key ‘vid’) and the coordinates of the vertices (key ‘xy’) was draw stored in ‘metadata’. The list in ‘xy’ in this application always started with the entry ‘7’ to mark the object as a polygon (https://gitlab.com/vgg/via/blob/master/via-3.x.y/src/js/_via_metadata.js,line15) [69]. The following entries need to be used in pairs, as *x*-coordinates and *y*-coordinates are interleaved. In the following example, three vertices are shown as interleaved *x*-, *y*-coordinates: (1.479, 23.671), (8.877, 11.836), and (2.959, 22.192).
“metadata”: {“1_pp7XiE0u”: {“vid”: “1”,“xy”: [7, 1.479, 23.671, 8.877, 11.836, 22.192, 1.479, *…*

#### 2.4.1. Training the Model

When annotation results were stored in distinct JSON files from various annotation sessions, the same image identification numbers would appear in different JSON files. In order to get a one-to-one correspondence between used images and identification numbers, the annotation results were fused assigning unique image identification numbers [71]. One JSON file for training and one JSON file for validation were created. Afterwards, the coordinates of the polygons’ vertices were loaded and were stored in the AxisCowDataset object and internally linked to the image identification.

### 2.5. Evaluating the Model: ‘Averaged Precision Score’ and ‘Averaged Recall Score’

The evaluation of the object detection model had to measure the performance regarding the existence of cows in the image. This was communicated by the model via confidence scores associated with every detection. In addition, the performance with regard to the location of the cow determined by the model needs to be evaluated, which was compared to the ground truth via the metric ‘Intersection over union’.

#### 2.5.1. Intersection over Union

The proportion between the intersection and the union of the ground truth bounding box and the bounding box output by the model is called ‘Intersection over union’ (IOU, Figure 4). It summarises how well the ground truth overlaps with the object predicted by the model. An IOU threshold 0.5 was set, which is a commonly used value [49,50,51] to mark a bounding box as a successful determination of the location of a cow. IOU can be calculated for the results of the segmentation mask prediction as well. In that case, the intersection and the union of the ground truth mask and the mask provided by the model were set into relation.

#### 2.5.2. Evaluation Metrics

The metrics’ true positives (*TP*), false positives (*FP*) and false negatives (*FN*) in object detection were determined according to Table 2. The confidence score *con_sc* was used as a measure, whether the model detected a cow. The ‘Intersection over Union’ (IOU) was used as measure, whether a cow was actually present at the anchor.

From these metrics, precision and recall (Equations (Equation 1) and (Equation 2)) were calculated. Precision is the percentage of correctly detected cows of all cows detected by the model. Recall counts the detected cows in relation to all cows present in the image material:(1)$$\mathrm{precision}\;=\;\frac{TP}{TP+FP}.$$
(2)$$\mathrm{recall}\;=\;\frac{TP}{TP+FN}.$$

##### Averaged Precision Score (AP) and Mean Averaged Precision Score (mAP)

Plotting the pairs (recall, precision) for a fixed IOU = 0.5 but varying thresholds T of *con_sc* (Table 2) gives the precision-recall curve pr. To calculate the metric ‘averaged precision score’ (AP), the precision values are averaged across a predefined set of recall values.
(3)$$\mathrm{AP}\;=\;\frac{1}{11}\times \sum_{r\in \{0,0.1,0.2,\ldots ,1\}}pr_{\mathrm{interp}}\left(r\right),$$
with $pr_{\mathrm{interp}}$ (Equation (Equation 4)) being an interpolation between the maxima in precision associated with recall values in {0,0.1,0.2,…,1} generated from the precision-recall curve pr:(4)$$pr_{\mathrm{interp}}\left(r\right)\;=\;\max_{r^{\prime }\geq r}\left\{p\left(r^{\prime }\right)\right\}\;\;;\;\;\;\;\;\;r\in \left\{0,0.1,0.2,\ldots ,1\right\}.$$

The mean of AP scores mAP for IOU values {0.5,0.55,…,0.95,1.0} is called ‘mean averaged precision score’.

##### Averaged Recall Score (AR)

Averaging the recall for the IOU values {0.5,0.55,…,0.95,1.0} calculates the metric ‘averaged recall score’ AR [48], which evaluates the performance of the model with respect to finding all present objects.

As overall evaluation metric mean(AP, mAP, AR) was calculated.

### 2.6. Exemplary Usage of Barn Space

The instance segmentation was applied to the video recordings made on 21 April 2019 to exemplarily determine the animals’ usage of the provided barn space. For this, the barn was partitioned into segments [21]. The lying cubicles were numbered ‘C01’ to ‘C36’ from left to right starting with the cubicles at the upper edge of the barn (Figure 1). Feeding and water troughs were numbered from ‘F01’ to ‘F18’, respectively, ‘D01’ and ‘D02’. The walking area was divided into 15 segments, named ‘X01’ to ‘X13’ as well as ‘L’ and ‘B’, whereby ‘L’ and ‘B’ were the segments where the licking stone and the cow brush was provided. For all segments, it was counted how many images of the video material the respective segment was occupied by at least one animal, i.e., a segmentation mask was shown in the segment.

### 2.7. Dependency on Size of the Training Data Set

To evaluate the dependency of the model performance on the number of images used for training, the model described in Section 2.3 was trained on only parts of the full training data set. Five repeated training runs were conducted, starting with twenty-one images for training and four images for validation to the full training data set, augmented by steps of twenty or twenty one, respectively, four or five images. In all runs, the used images were randomly chosen from the full training data set. The models were trained for 350 epochs and saved after each epoch. Afterwards, the weights from the epoch with minimal validation loss were chosen. These weights were applied on the full validation data set to calculate AP, mAP, AR and mean(AP, mAP, AR) (Section 2.5) for both masks and bounding boxes as measures for model performance.

In order to test for the effect of the number of images, Kruskal–Wallis tests were performed on the evaluation metrics AP, mAP, AR and mean(AP, mAP, AR) grouped after the used size of the training data set. Furthermore, the minimal validation loss during training and the proportion of training time, i.e., the number of trained epochs divided by 350, were grouped after the used size of the training data set, and Kruskal-Wallis tests were conducted.

## 3. Results

### 3.1. Segmentation of Cows

Figure 5 shows segmentation results achieved by the trained Mask R-CNN model applied to data which was neither part of training nor the validation data set.

The evaluation metrics of the model trained on the full data set can be found in Table 3 in comparison with the performance of the Mask R-CNN model with ResNet50+FPN backbone on the MS COCO data as reported by [51].

### 3.2. Exemplary Usage of Barn Space

Figure 6 represents the animals’ usage of the provided barn space as automatically determined by the Mask R-CNN instance segmentation exemplarily for 21 April 2019. The mostly used barn areas are in the running area, especially in the upper part of the barn. The lying cubicles in the center of the barn have been used more frequently than the lying cubicles at the upper edge of the barn.

### 3.3. Dependency on the Size of the Training Data Set

In Figure 7, the boxplots on basis of grouping after the used proportion of the training data set are illustrated. Kruskal-Wallis tests revealed significant influence (*p* = 0.01) of the size of the used training data set on validation loss, optimal stopping time, and the evaluation metrics mAP, AR and mean (AP, mAP, AR). The ‘averaged precision score’ AP did not vary significantly with the size of training data set.

## 4. Discussion

In this article, loose-housed dairy cows were segmented successfully from video material using the cutting edge deep learning model Mask R-CNN. The modifications made to the implementation by [52] prior to transfer learning and the preparation of the training data set with VGG Image Annotator [70] were explained in detail to enable straightforward applications in further scientific studies. The respective code for transfer learning, for the handling of the JSON files generated during annotation and for the calculation of evaluation methods can be found on GitHub [71].

Recently, Ref. [54] has once more proven that Mask R-CNN is a superior model when it comes to instance segmentation, while comparing the method to other state-of-the-art object detection models. Their application was based on video material of cattle in outdoor environments (pasture, feedlock). The present study focusses on an indoor environment, which faces different challenges regarding light conditions, dirt, humidity and, thus, lower contrast between animals and background. Compared to the pasture, partial occlusions of cows happen more often due to the stronger restriction of space. Furthermore, the presence of installations such as key pillars, fans, lying cubicles and automated manure scraper cause additional occlusions and complicate image annotation. Another difference lies in the source of video material. While in the outdoor setting Ref. [54] could record cattle by a quadcopter vision system, the presented setting had to rely on multiple firmly installed surveillance cameras to cover the complete barn area. The complete surveillance of the cow barn featuring multiple surveillance cameras was deployed as a new setting holding various technical and analytical challenges. With this, our model on the one hand also needed to detect incomplete animals, as cows could leave the field of view of individual cameras and appear in the neighbouring field of view. On the other hand, barrel distortion was visible towards the edges of the cameras’ fields of view. This led to significantly smaller cow representations at the image boundaries compared to the cows depicted in the image centre. Visual inspections of the results showed that the Mask R-CNN model performed successfully on this task as well (Figure 5). The surveillance cameras were mounted in 3.5 m height and along the centre line of the barn segment to avoid manipulation by the animals. It would have been beneficial to mount the cameras in two rows parallel to the centre line to reduce barrel distortion at the edges of the barn area. However, this was not feasible, firstly due to an overload in data traffic caused by additional cameras, and secondly due to constructional reasons with regard to the participating farm.

The Mask R-CNN implementation of [52] supports the backbone network architectures ResNet50 and ResNet101. Here, ResNet50 was chosen over ResNet101, as deeper models are more difficult to train, and the additional training data set was comparably small. Furthermore, using the deeper network ResNet101 as backbone would have significantly increased computational costs, and it was necessary that training was not performed on high-end multi-GPU systems but on more affordable hardware. To generate training and validation data set, images were randomly chosen from recording days evenly distributed over the complete data collection period. The reasons for this method of choice were that images from different times of the day but also different months were needed for training in order to enhance the generalisability of the model. Furthermore, as the cows did not change position rapidly, the images that were taken within a singled out recording hour would have been too similar to be used for training a successful network. Transfer learning [72] was applied using the weights trained on the MS COCO data set as start values instead of starting with randomly assigned weights. The MS COCO data set contains approximately 120,000 images. During pre-training, the model learned a lot of features that are common in natural images and was even trained to detect the object class ‘cow’. As the model is in this way provided with a priori knowledge in detecting and segmenting various objects and animals in the respective natural context, the training time can be reduced. Regarding the decision on which layers of the model should be trained further, and for which layers the parameters are left as in the pre-trained condition, the size of the training data set matters as well as its similarity compared to the MS COCO data set [73]. Even the full training data set used in this study (479 images for one class) has to be considered small. However, as the MS COCO data set also contains a class ‘cow’ in addition to various other farm animals, a medium degree of similarity between the data sets could be assumed. With close similarity, it would have been sufficient to only train the heads of the model, i.e., the RPN and the three output branches. Training additional ResNet50 stages could enhance the performance of the model, but also increases the risk of overfitting. Preliminary tests (results not presented) showed that the best results were provided by a model trained from ResNet50 stage four upwards.

The performance measures used in this study were based on counting true positive, false positive and false negative detections of cows. A true negative detection would refer to the situation where there is no cow to detect, i.e., background, and the model has output a confidence score smaller than 0.9. These situations are not meaningfully countable, as already the RPN determines the objectness, and only anchors with an object present were transferred to the following stages and assigned a confidence score. The calculated evaluation metric ‘averaged precision score’ (AP) for an IOU threshold of 0.5 has become widely accepted [55,74,75] in the evaluation of object detection models. With 0.91 and 0.85 ‘averaged precision score’ for the bounding box determination and the segmentation masks, respectively, the Mask R-CNN model performed extremely well. Strongly emphasising the exact overlapping of ground truth and detection, in the calculation of the ‘mean averaged precision score’ (mAP) and the ‘averaged recall score’ (AR) the AP and the recall (Equation (Equation 2)) were averaged for IOU thresholds from 0.5 to 0.95. For the model trained for this study, even these evaluation metrics, which demand a very precise determination of the cows’ positions, could be considered moderate to high. The full training data set used in this analysis was very small compared to publicly available datasets which include several object classes as PASCAL VOC, ILSVRC or MS COCO [46,47,48,74]. However, transfer learning cows were later successfully segmented (Figure 5), and evaluation metrics compared to the pre-trained model (Table 3) were clearly improved.

Deep CNN models for object detection have already been applied in animal related science [42,44,54] and multiple additional applications are imaginable. However, to enhance the performance of a pre-trained model, it is advantageous to generate a specific training data set for the respective setting and desired application. This is extremely time consuming and tedious, thus the question “How many images do I need to annotate?” is of great interest. In this article, model performance was additionally evaluated for increasing the size of the training data set. The respective results show a significant decrease in validation loss as well as a significant increase in the overall model performance mean(AP, mAP, AR) for both bounding boxes and segmentation masks, but also an increase in computation time till optimal stopping. It has to be concluded that for a pre-trained network even the annotation of 50 or 100 additional images for transfer learning could enhance the aspired object detection considerably. This might be an important finding for the conduction of CNN based studies, when the human effort or time are limited resources.

Information on the daily activities of the herd could also be derived by visually inspecting the video material. However, the machine learning approach presented in this article provides the possibility to use the complete material of longterm recordings to analyse herd activity, which would of course not be feasible in a visual inspection. To name possible applications, an automated determination of the usage of different areas of the provided space [76] is supported. The results shown in Figure 6 should not be interpreted scientifically, as they are only based on one day, but they serve as an example for the applicability of the provided instance segmentation for analyses regarding the herd activities. In comparison to the motion detection used in [21], basing the analysis of space usage on instance segmentation comes with the additional opportunity to determine not only how often an area is used, but to follow the course of the individual instances once object tracking has been implemented. Instance segmentation results could also be considered more valuable as bounding box object detection, as a more precise distinction between occluded animals and the orientation of the animals could be derived from the masks, without an additional model determining the head or meaningful keypoints [54]. An analysis of a more specific use of resources [77] becomes thinkable, as it might exemplarily be possible to automatically determine whether the cow is actually using the cow brush or just standing next to it. Concerning the interactions between animals, the distinction between head to head contacts from head to rump or head to pelvis contacts becomes within reach. Future work is going to focus on the implementation of object tracking for which the detection of individual instances is the basis. This would for example enable an automated determination of contact networks and a description of the group structure. The present work thus provides solid technical groundwork for these important applications with regard to animal behaviour and husbandry.

## 5. Conclusions

In this study, a deep learning Mask R-CNN model was applied to detect loosed-housed Holstein Friesian dairy cows from the video material of firmly installed surveillance cameras. Transfer learning was applied, while the starting weights were derived by pre-training the model on the Microsoft common objects in the context data set. The animals were successfully segmented reaching high ‘averaged precision scores’ for bounding boxes (0.91) and segmentation masks (0.85) for a given IOU threshold of 0.5. The more strict performance measures ‘mean averaged precision score’ and ‘averaged recall score’ could be considered moderate to high. In addition, the effect of transfer learning and the size of the additionally necessary training data set were made countable. As cattle are a highly sociable species, the interaction in the herd structure has a massive impact on animal welfare. The article presents the technical basis to automatically track cows in the video material, as individual instances of animals were determined. Thus, groundwork was provided for automated analyses of herd activity in loose-housing as well as the use of resources.

## Figures and Tables

**Figure 1 animals-10-02402-f001:**
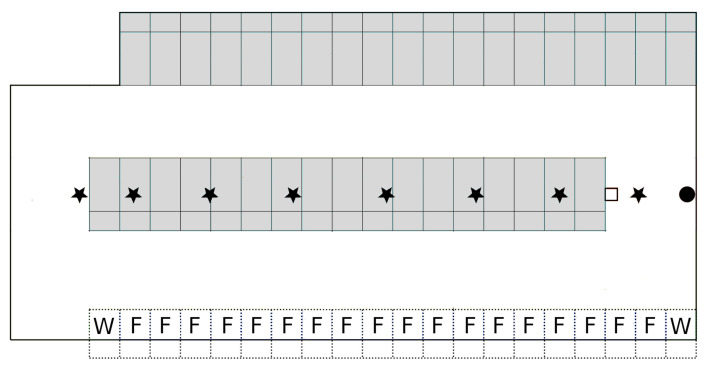
Schematic representation of the barn. The lying cubicles were highlighted grey. The water and feeding troughs are denoted with ‘W’ and ‘F’, respectively. The camera positions are marked by black stars. The licking stone and the cow brush are depicted as white square and black dot, respectively.

**Figure 2 animals-10-02402-f002:**
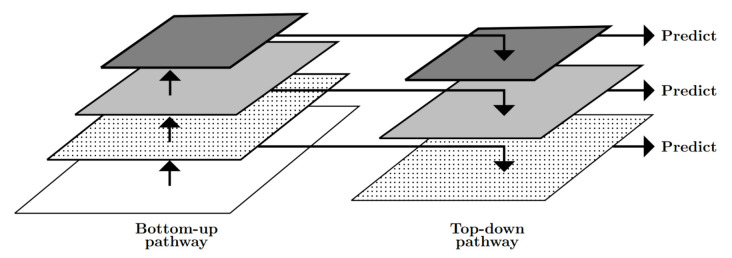
Schematic representation of the Feature Pyramid Network (FPN).

**Figure 3 animals-10-02402-f003:**
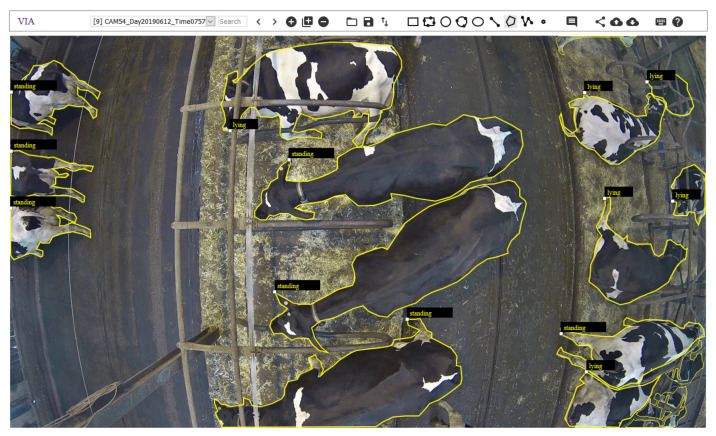
Example of an image that was manually annotated with VIA 3—apart from the outlines of the cows it was labelled if the animal was ‘standing’ or ‘lying’. The posture information has not yet been used for training the model.

**Figure 4 animals-10-02402-f004:**
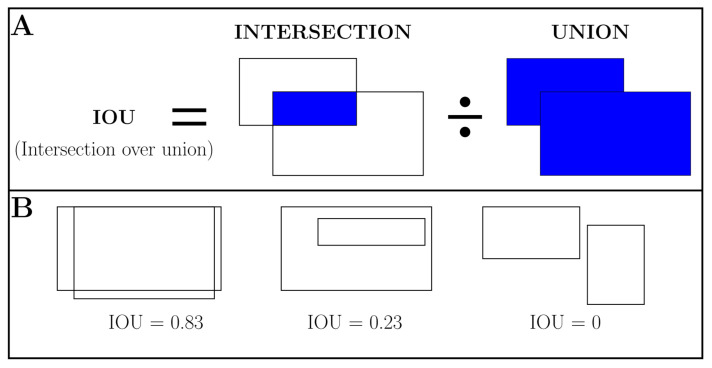
Intersection over union (IOU) exemplarily for bounding boxes. (**A**) The IOU is calculated by dividing the intersection of the two boxes by the union of the boxes; (**B**) examples for different values of IOU.

**Figure 5 animals-10-02402-f005:**
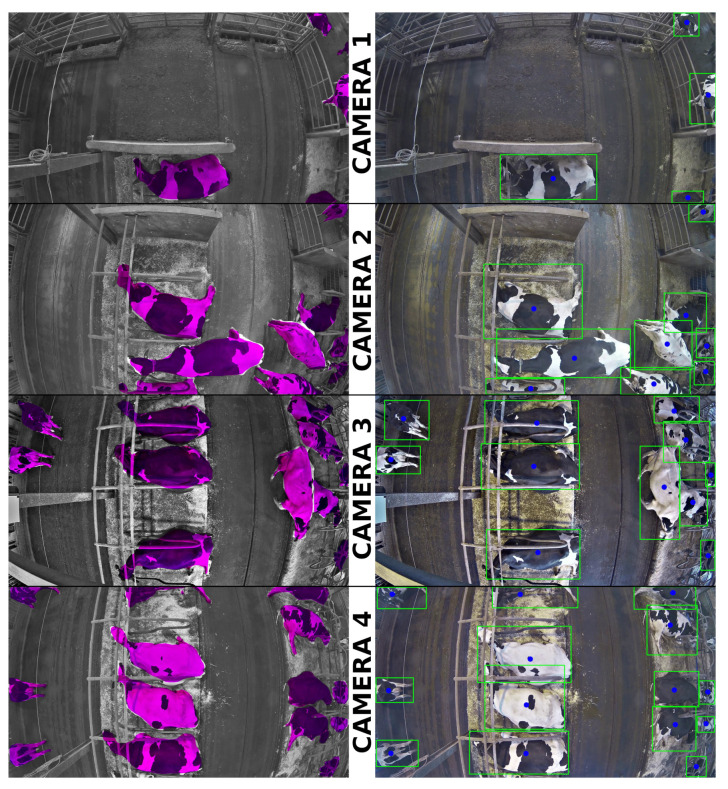
Examples of the resulting cow detection on unseen data for cameras 1 to 4. On the left, the masks are visualized. On the right, the bounding boxes are displayed. The centroids of the masks are marked blue.

**Figure 6 animals-10-02402-f006:**
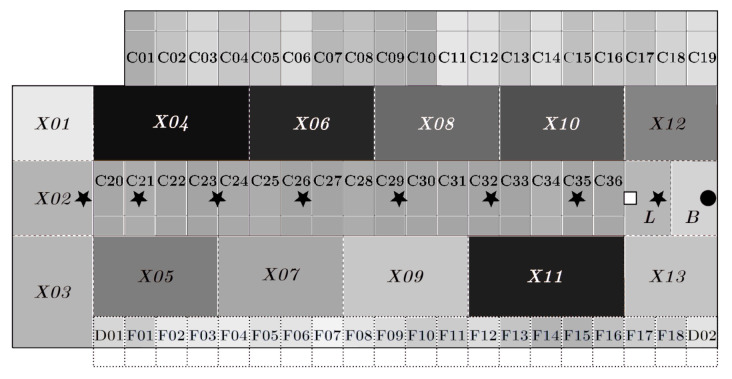
Visualisation of the usage of barn space on the example day 21 April 2019. The darker the barn segment depicted, the more time it was occupied by cows.

**Figure 7 animals-10-02402-f007:**
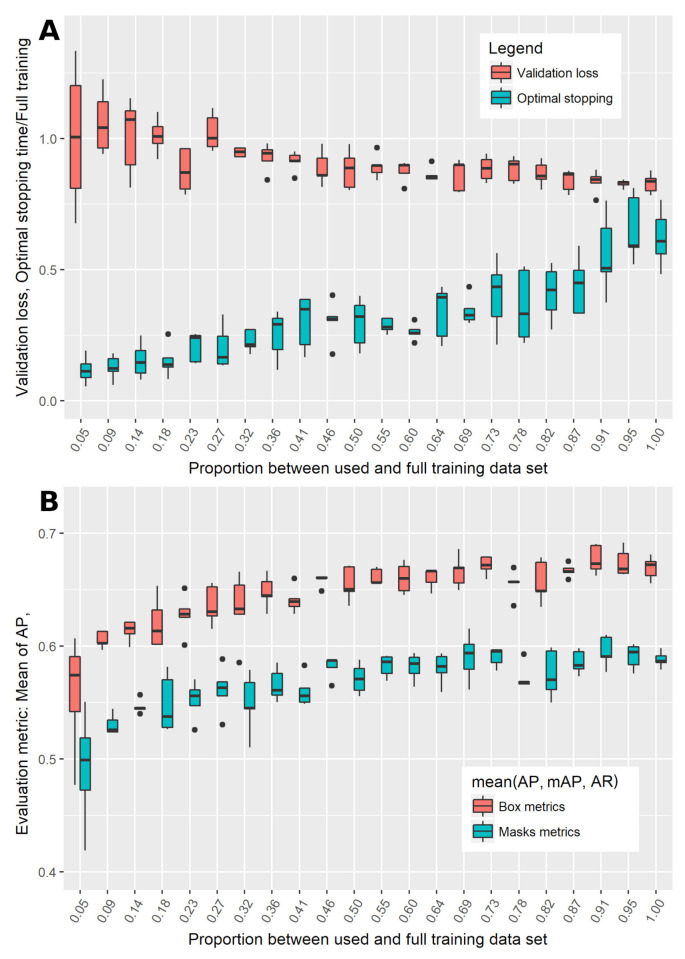
Boxplots for the Mask R-CNN performance depending on the size of the training data set. (**A**) validation loss and time for optimal stopping of model training; (**B**) mean of ‘averaged precision score’ (AP), ‘mean averaged precision score’ (mAP) and ‘averaged recall score’ (AR), mean(AP, mAP, AR) for the detection of bounding boxes and segmentation masks.

**Table 1 animals-10-02402-t001:** Number of images used for training and validation data set for all eight cameras.

Camera	Images for Training	Images for Validation
CAM 1	58	11
CAM 2	60	12
CAM 3	59	12
CAM 4	62	12
CAM 5	62	13
CAM 6	61	13
CAM 7	60	12
CAM 8	57	11
∑	479	96

**Table 2 animals-10-02402-t002:** Definition of true positives (*TP*), false positives (*FP*) and false negatives (*FN*) in object detection. *TP*, *FP* and *FN* were calculated for varying thresholds ‘T’ of the confidence score *con_sc*, a measure of whether the model detected a cow. IOU denotes the ‘Intersection over union’, a measure of whether a cow was actually present.

Model Output Compared to Ground Truth		Evaluation
con_sc≥ T (Cow detected)	IOU >0.5 (Cow present)	*TP*
	IOU ≤0.5 (No cow present)	*FP*
Missed cow: con_sc< T, but a cow was present at the anchor		*FN*

**Table 3 animals-10-02402-t003:** Evaluation metrics ‘averaged precision score’ (AP), ‘mean averaged precision score’ (mAP) and ‘averaged recall score’ (AR) of the fully trained model for detection of bounding boxes and segmentation masks. The third column contains the performance of the Mask R-CNN model with ResNet50-FPN backbone on the COCO data.

Metric	Bounding Box	Segmentation Mask	COCO (Masks)
AP	0.909	0.849	0.552
mAP	0.584	0.454	0.336
AR	0.606	0.551	not reported

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
