# Peer review of "Instance Segmentation with Mask R-CNN Applied to Loose-Housed Dairy Cows in a Multi-Camera Setting"

_animals, 2020, doi:10.3390/ani10122402_

Round 1
Reviewer 1 Report
Dear authors,
General comments:
The article presents a machine learning approach to detect cows in different parts of a barn. The topic is interesting, and the article is mostly well written. Nevertheless, a high level of detail regarding the machine learning techniques likely makes it a bit challenging to read for people within animal sciences. In the Simple Summary and Abstract it is emphasized how herd activity is an important indicator of the health and welfare status of the flock. However, this is not followed up on in the rest of the paper apart from a small section on future perspectives at the end of the discussion. I miss the link between the presented results (evaluation metrics) and how the model can be used to inform the farmer. Can the authors in some way link the results to the presence of the cows? I understand that this is “technical groundwork”, however the farmer can look at the camera image and quickly assess were the majority of his herd is, so what more does the machine learning provide?
Another key issue is that the paper lacks a clear aim. Which consequently makes it a bit challenging to appraise the results section as one does not quite know what to expect.
Specific comments
Introduction
A major point regarding the introduction is that it seems to be a mix of background information and a description of the methods used in the paper. This needs to be sorted out.
L44: “As cattle would naturally live in herds and have a strong need for companionship, dairy cows were usually kept in groups.” Please revise language, consider using present tense.
L 50: “…but housing and management influence the welfare status more than the herd size, although most groups on commercial dairy farms are unnaturally large.” What is unnaturally large? Compared to what? Flock size of ruminants in the wild? I suggest removing this statement.
L55: “Visual observations as in the analysis of the synchronisation of herd activity presented by [28], but also visual observation combined with positioning sensors ([29]) were used for data collection.” In what study was visual observations used? This study? In that case this is m&m and should be moved accordingly.
L59: “Hereby travelling distances ([30]), walking behaviour ([31]) as well as social networks and group structure ([32]) were dealt with.” Again, in what study was this dealt with? If referring to the present study move to m&m section.
L60: “It could not be totally excluded, that the animals were disturbed by the attached sensors. Additionally, the wireless sensors were prone to destruction by the animals as well as interferences with humidity or metal ([33]). Also, the necessary trade-off between data sampling frequency and battery lifetime could significantly affect the results of the study ([30]).” If this refers to the present study it is a consideration of methods and potential weaknesses and should be moved to the discussion.
L64-64: This also sounds like methods? If the purpose is to introduce the reader to general concepts, it needs to be reformulated in a way that makes this clear.
L72-77. “As full connected neural networks… … to accelerate the training ([38]; [39]). Most of this seems like description of the methods used.
L99-115: This section also seems to be a mix of introducing the reader to relevant concepts and a description of methods.
L155-117. This is were I would expect to see the aim of the study, but what is written here reads more like a conclusion.
L274-275: “An IOU threshold 0.5 was set, which is a commonly used value to mark a bounding box as a successful determination of the location of a cow.” Could a reference be provided?
L302: Why is a hypothesis test performed? What answer does this p-value provide?
Materials and methods
Well-structured with nice figures.
Table 2. Suggest having an explanation for con_sc in the Table caption.
L150-151: I assume it is mean lactation and SEM that is provided in parentheses? Please write this explicitly. I would consider using the median and IQR (or range) instead given that lactation number is often skewed, however as I have not seen the distribution, I cannot say for the present study.
L151-152: “Daily averaged milk yields varied from 22.3 kg to 50.1 kg (36.2 kg ± 5.8 kg).” Averaged over what? Weeks, months?
Consider adding information about milking system (parlour/AMS/pipeline?)
Results
After seeing Table 2. in the m&m section I would expect a table of counts (TP, FP and FN) in the results section, from which the performance measures were based on.
The results section is very short. Is there nothing more to provide? As mentioned under general comments I would appreciate some results that says something about the cows, not just model performance.
Other comments:
I notice all citations only provides the number of the citation, even if the citation is included as a subject in a sentence.
E.g. “The problem of body condition determination was successfully approached by [10] using thermal cameras,…”
I would expect this to read “The problem of body condition determination was successfully approached by Halachmi et al. [10] using thermal cameras,…”
I do not know if this has something to do with the reference style of the journal and leave this to the editorial
Author Response
Dear Mrs., Mr,
Thank you for reviewing our article named ’Instance segmentation with Mask R-CNN applied to loosehoused dairy cows in a multi-camera setting’. I have reviewed your remarks and tried to make the paper stronger by taking into account your constructive comments on this paper. I am sure these comments improve the quality of the paper and I hope they meet your expectations. Below you can see my answers (italics) to your comments. The adjustments I made are highlighted in the resubmitted manuscript. If there are any questions about them, I will be glad to answer them.
1. Comments and Suggestions for Authors Dear authors, General comments: The article presents a machine learning approach to detect cows in different parts of a barn. The topic is interesting, and the article is mostly well written. Nevertheless, a high level of detail regarding the machine learning techniques likely makes it a bit challenging to read for people within animal sciences. In the Simple Summary
and Abstract it is emphasized how herd activity is an important indicator of the health and welfare status of the flock. However, this is not followed up on in the rest of the paper apart from a small section on future perspectives at the end of the discussion. I miss the link between the presented results (evaluation metrics) and how the model can be used to inform the farmer. Can the authors in some way link the results to the presence of the cows? I understand that this is ?technical groundwork?, however the farmer can look at the camera image and quickly assess were the majority of his herd is, so what more does the machine learning provide?
This reviewer is correct, that the farmer could look at the camera recordings to see, where his herd resides. But to monitor the preferred areas of the barn or the use of resources by looking at the screen is not feasible, as the farmers’ time is too precious for that. A machine learning approach, however, could collect this information and provide insights automatically without the need to invest valuable time in front of the screen. The instance segmentation presented here might even be used to automate the detection of social contacts which is still in development, thus, it is not yet presented. Using not this CNN based instance segmentation, but a more rough motion detection approach to localise the animals, the authors have already published articles on the usage of barn space and resources. We considered it double information to put results of the same type but based on the new technical groundwork into this article. As this reviewer suggests, the machine learning approach needs an application to be valued and understood properly, we now visualized the space usage pattern based on the instance segmentation for
an example day as additional results. Please compare also our answer to your comment 19.
2. Another key issue is that the paper lacks a clear aim. Which consequently makes it a bit challenging to appraise the results section as one does not quite know what to expect.
Thank you for pointing this out. Compare our answer to your valuable comment 10.: The last passage of the introduction was rephrased to clarify th aim of the article.
3. Specific comments Introduction A major point regarding the introduction is that it seems to be a mix of background information and a description of the methods used in the paper. This needs to be sorted out. L44: ?As cattle would naturally live in herds and have a strong need for companionship, dairy cows were usually kept in groups.? Please revise language, consider using present tense.
Thank you. The sentence was switched to present tense.
4. L 50: ??but housing and management influence the welfare status more than the herd size, although most groups on commercial dairy farms are unnaturally large.? What is unnaturally large? Compared to what? Flock size of ruminants in the wild? I suggest removing this statement.
Thank you for your suggestion. The statement was removed.
5. L55: ?Visual observations as in the analysis of the synchronisation of herd activity presented by [28], but also visual observation combined with positioning sensors ([29]) were used for data collection.? In what study was visual observations used? This study? In that case this is m&m and should be moved
accordingly.
The Visual observations have taken place in the cited study [28] and the combination of visual observation and positioning sensors has taken place in the other cited study [29], with
28. Šárová, R.; Špinka, M.; Panamá, J.L.A. Synchronization and leadership in switches between resting and activity in a beef cattle herd – A case study. Applied Animal Behaviour Science 2007, 108, 327 - 331. doi:10.1016/j.applanim.2007.01.009.
and
29. Nelson, S.; Haadem, C.; Nødtvedt, A.N.; Hessle, A.; Martin, A. Automated activity monitoring and visual observation of estrus in a herd of loose housed Hereford cattle: Diagnostic accuracy and time to ovulation. Theriogenology 2017, 87, 205 - 11. doi:j.theriogenology.2016.08.025.
The sentence was rephrased to make this more clear.
6. L59: ?Hereby travelling distances ([30]), walking behaviour ([31]) as well as social networks and group structure ([32]) were dealt with.? Again, in what study was this dealt with? If referring to the present study move to m&m section.
Travelling distances, walking behaviour and social networks/group structure were analysed in the sensor based studies [30], [31], and [32], with
Davis, J.; Darr, M.; Xin, H.; Harmon, J.; Russell, J. Development of a GPS Herd Activity and Well- Being Kit (GPS HAWK) to Monitor Cattle Behavior and the Effect of Sample Interval on Travel Distance. Applied Engineering in Agriculture 2011, 27. doi:10.13031/2013.36224,
Rose, T. Real-time location system Series 7000 from Ubisense for behavioural analysis in dairy cows. PhD thesis, Institute of Animal Breeding and Husbandry, Kiel University, 2015.
and
Boyland, N.K.; Mlynski, D.T.; James, R.; Brent, L.J.N.; Croft, D.P. The social network structure of a dynamic group of dairy cows: From individual to group level patterns. Applied Animal Behaviour Science 2016, 174, 1 - 10. doi:10.1016/j.applanim.2015.11.016.
The sentence was rephrased to make this more clear.
7. L60: ?It could not be totally excluded, that the animals were disturbed by the attached sensors. Additionally, the wireless sensors were prone to destruction by the animals as well as interferences with humidity or metal ([33]). Also, the necessary trade-off between data sampling frequency and battery lifetime could significantly affect the results of the study ([30]).? If this refers to the present study it is a consideration of methods and potential weaknesses and should be moved to the discussion.
This passage does not refer to the present study, but provides the disadvantages of the use of wireless sensors in animal based studies to the reader as important background information backed up by references. The authors rephrased this passage.
8. L64-64: This also sounds like methods? If the purpose is to introduce the reader to general concepts, it needs to be reformulated in a way that makes this clear.
In this passage, the advantageous fact, that the data from wireless sensors need less pre-processing than image data is stated as important background information. It does not refer to the present study, as no wireless sensors are used in this study. The sentence was rephrased.
9. L72-77. ?As full connected neural networks? ? to accelerate the training ([38]; [39]). Most of this seems like description of the methods used.
The authors have not used fully connected neural networks in the present paper, thus, this is no description of the methods used. Instead this is again meant to provide the reader appropriately with background and references. The authors considered historical information on CNNs as given in this passage relevant
for the understanding of the background regarding this method and why it has not been applied directly after its introduction. However, in Materials and methods the text would have been burdened by this information. The passage was rephrased.
10. L99-115: This section also seems to be a mix of introducing the reader to relevant concepts and a description of methods.
Thank you for pointing this out, however, the authors consider the passage between ’In the present article, a Mask R-CNN model (Mask region-based CNN’ and ’learning to generate a training data set from the respective setting’ to be strictly the provision of background information, as general knowledge on the
used model and transfer learning was formulated together with citing relevant literature.
Regarding the rest of the passage the reviewer is referring to in this comment and the rest of the section ’Introduction’: The authors rephrased it, to formulate the aim of the study more clearly.
11. L155-117. This is were I would expect to see the aim of the study, but what is written here reads more like a conclusion.
See our answer to your valuable comment 10.
12. L274-275: ?An IOU threshold 0.5 was set, which is a commonly used value to mark a bounding box as a successful determination of the location of a cow.? Could a reference be provided?
References were provided.
13. L302: Why is a hypothesis test performed? What answer does this p-value provide?
The Kruskal Wallis tests were performed to test for the effect of the number of images in the training data set. This additional explanation was added to the text. As the size of the training data set affects validation loss, optimal stopping time, and the evaluation metrics mAP, AR and mean(AP, mAP, AR), however, NOT ’averaged precision score’ AP, conducting the test was meaningful.
14. Materials and methods Well-structured with nice figures. Table 2. Suggest having an explanation for con_sc in the Table caption.
Thank you for this suggestion. We added explanation to the caption. Please compare also our answer to your valuable comment 18.
15. L150-151: I assume it is mean lactation and SEM that is provided in parentheses? Please write this explicitly. I would consider using the median and IQR (or range) instead given that lactation number is often skewed, however as I have not seen the distribution, I cannot say for the present study.
Thank you for pointing this out. The mean and standard deviation were exchanged in favour of the median. The range has already been given in the worded sentence.
16. L151-152: ?Daily averaged milk yields varied from 22.3 kg to 50.1 kg (36.2 kg +- 5.8 kg).? Averaged over what? Weeks, months?
This reviewer is right, an error has occurred here, as due to drying of, the group of animals could not be kept constant over the complete recording period. More detailed information on the cows have been provided regarding the time frame between animal exchanges. The authors hope that the animals are
now adequately described.
17. Consider adding information about milking system (parlour/AMS/pipeline?)
Thank you for this suggestion. The information was added.
18. Results After seeing Table 2. in the m&m section I would expect a table of counts (TP, FP and FN) in the results section, from which the performance measures were based on.
The ’averaged precision score’ AP is calculated as average over a set of precision values, but each precision value is calculated from individual TP, FP and FN values for varying thresholds of con_sc. Thus, there is not the one specific combination of TP, FP and FN values behind the AP metric (or the other used object detection metrics used here), but a set of combinations for varying con_sc thresholds. The authors consider it not helpful to provide this large amount of TP, FP and FN values. Especially, as the metrics provided were the usually reported metrics in object detection. The authors tried to make this more clear by rephrasing the sentence and introducing the con_sc threshold T in Table 2 where TP, FP and FN
were declared.
19. The results section is very short. Is there nothing more to provide? As mentioned under general comments I would appreciate some results that says something about the cows, not just model performance.
This work was meant to provide technical groundwork for various future analyses aimed at the cows, i.e automated detection of contacts between the animals. These detection has not been developed yet and is, thus, not presented in this work; please compare our answer to your comment 1. . However, the authors
followed the this reviewer’s suggestion to enhance the section ’Results’ and to motivate the usage of the machine learning approach presented here, and we added a visualisation of space usage pattern of an example day as additional results based on the here presented instance segmentation.
20. Other comments: I notice all citations only provides the number of the citation, even if the citation is included as a subject in a sentence. E.g. “The problem of body condition determination was successfully approached by [10] using thermal cameras,...” I would expect this to read “The problem of body condition determination was successfully approached by Halachmi et al. [10] using thermal cameras,...” I do not know if this has something to do with the reference style of the journal and leave this to the editorial
The authors used the MDPI Latex template to typeset this manuscript. As a matter of fact, the form of references was determined by this template.

Reviewer 2 Report
The solution proposed is to use multi-cameras in order to acquire information about the posture and position of individual cows in the natural environment. This is a very challenging problem because the cows tend to be in groups, and is very common to have a superposition of images of the cows in the video frame.
The introduction address properly the main topics of the paper, with enough context supported by bibliographic analysis of the state of the art. But, I did not understand the objective expressed for this paper. Please, clarify the objective of the paper.
The size of the sample is big enough for the purpose of the paper. The authors use 8 cameras, with 575 images for training and validation. I understand that those images have differences between them (for example, are they a sequence of photos taken from the video cameras or they are taken at different time instants).
The conclusions are supported by the results, but they have space to elaborate. It is not clear why the presented solution contributes to solving the problem addressed in the introduction. For example, the affirmation made about "the presented results provide the technical groundwork for automated analyses of herd activity in loose-housing as well as the use of resources". need to be supported by an explanation based on the results achieved in this paper.
Can the proposed methodology be used for tracking the movements of cows? This is to clarify how this method should be used to identify the activity of the cows.
Author Response
Dear Mrs., Mr,
Thank you for reviewing our article named ’Instance segmentation with Mask R-CNN applied to loose housed dairy cows in a multi-camera setting’. I have reviewed your remarks and tried to make the paper stronger by taking into account your constructive comments on this paper. I am sure these comments improve the quality of the paper and I hope they meet your expectations. Below you can see my answers (italics) to your comments. The adjustments I made are highlighted in the resubmitted manuscript. If there are any questions about them, I will be glad to answer them.
1. Comments and Suggestions for Authors The solution proposed is to use multi-cameras in order to acquire information about the posture and position of individual cows in the natural environment. This is a very challenging problem because the cows tend to be in groups, and is very common to have a
superposition of images of the cows in the video frame.
The introduction address properly the main topics of the paper, with enough context supported by bibliographic analysis of the state of the art. But, I did not understand the objective expressed for this paper. Please, clarify the objective of the paper.
Thank you for your valuable comment. The authors rephrased the last passage of the section ’Introduction’ to make the objective of the article more clear.
2. The size of the sample is big enough for the purpose of the paper. The authors use 8 cameras, with 575 images for training and validation. I understand that those images have differences between them (for example, are they a sequence of photos taken from the video cameras or they are taken at different time instants).
This is correct.
3. The conclusions are supported by the results, but they have space to elaborate. It is not clear why the presented solution contributes to solving the problem addressed in the introduction. For example, the affirmation made about "the presented results provide the technical groundwork for automated analyses
of herd activity in loose-housing as well as the use of resources". need to be supported by an explanation based on the results achieved in this paper.
Thank you for pointing this out. The authors reworked the section ’Conclusion’.
4. Can the proposed methodology be used for tracking the movements of cows? This is to clarify how this method should be used to identify the activity of the cows.
This reviewer is right. The proposed methodology lays the basis for tracking the movement of the cows. The authors tried to clarify this matter.

Round 2
Reviewer 1 Report
The Authors have addressed most of my concerns.
Regarding in text citations I recommend double checking the style. MDPI Reference guide refers to the ASC Style Guide. In the manuscript, the current format of in text citations is not in line with ASC guidelines;
eg.
"[22] showed that the social structure in herds of semi-wild
cattle was based on matriarchal families,..."
Should read:
"Reinhart et al. [22] showed that the social structure in herds of semi-wild
cattle was based on matriarchal families,..."
I leave this issue to the editorial team.
Reviewer 2 Report
Congratulations on your hard work. The manuscript has improved after the first review, and it solved all my doubts. I have no more comments to add to this second revision.
best regards.